# Synergistic and Detrimental Effects of Alcohol Intake on Progression of Liver Steatosis

**DOI:** 10.3390/ijms23052636

**Published:** 2022-02-27

**Authors:** Agostino Di Ciaula, Leonilde Bonfrate, Marcin Krawczyk, Gema Frühbeck, Piero Portincasa

**Affiliations:** 1Clinica Medica “Augusto Murri”, Department of Biomedical Sciences and Human Oncology, University of Bari Medical School—Piazza Giulio Cesare 11, 70124 Bari, Italy; agostinodiciaula@tiscali.it (A.D.C.); leonilde.bonfrate@uniba.it (L.B.); 2Department of Medicine II Saarland University Medical Center, Saarland University, 66424 Homburg, Germany; marcin.krawczyk@uks.eu; 3Laboratory of Metabolic Liver Diseases, Department of General, Transplant and Liver Surgery, Centre for Preclinical Research, Medical University of Warsaw, 02-091 Warsaw, Poland; 4Department of Endocrinology & Nutrition, Clínica Universidad de Navarra, 31008 Pamplona, Spain; gfruhbeck@unav.es; 5Metabolic Research Laboratory, Clínica Universidad de Navarra, 31008 Pamplona, Spain; 6CIBER Fisiopatología de la Obesidad y Nutrición (CIBEROBN), ISCIII, 31009 Pamplona, Spain; 7Obesity and Adipobiology Group, Instituto de Investigación Sanitaria de Navarra (IdiSNA), 31009 Pamplona, Spain

**Keywords:** liver steatosis, liver cirrhosis, binge drinking

## Abstract

Nonalcoholic fatty liver disease (NAFLD) and alcoholic liver disease (ALD) are the most common liver disorders worldwide and the major causes of non-viral liver cirrhosis in the general population. In NAFLD, metabolic abnormalities, obesity, and metabolic syndrome are the driving factors for liver damage with no or minimal alcohol consumption. ALD refers to liver damage caused by excess alcohol intake in individuals drinking more than 5 to 10 daily units for years. Although NAFLD and ALD are nosologically considered two distinct entities, they show a continuum and exert synergistic effects on the progression toward liver cirrhosis. The current view is that low alcohol use might also increase the risk of advanced clinical liver disease in NAFLD, whereas metabolic factors increase the risk of cirrhosis among alcohol risk drinkers. Therefore, special interest is now addressed to individuals with metabolic abnormalities who consume small amounts of alcohol or who binge drink, for the role of light-to-moderate alcohol use in fibrosis progression and clinical severity of the liver disease. Evidence shows that in the presence of NAFLD, there is no liver-safe limit of alcohol intake. We discuss the epidemiological and clinical features of NAFLD/ALD, aspects of alcohol metabolism, and mechanisms of damage concerning steatosis, fibrosis, cumulative effects, and deleterious consequences which include hepatocellular carcinoma.

## 1. Introduction

Chronic liver diseases (CLDs) have a major impact on health care systems worldwide. Liver cirrhosis represents the 11th leading cause of death, and hepatocellular carcinoma (HCC) is the 4th leading cause of cancer death. [1]. The mortality trends from both CLDs and HCC have been increasing in several countries [2,3]. In addition, liver cirrhosis is characterized by a substantial economic burden and is one of the top 20 causes of disability-adjusted life years [1]. Among CLDs, both nonalcoholic fatty liver disease (NAFLD) and alcohol-related liver disease (ALD) remain the two main liver diseases worldwide [1]. In the general population, NAFLD and ALD show high prevalence, and trends are increasing. Both NAFLD and ALD share many similarities and, at low alcohol consumption, often coexist. Recent observations point to the synergistic effects of alcohol and liver steatosis in driving liver damage. 

In this review, we discuss the overall pathophysiological pathways involved in ALD and NAFLD to understand how the ultimate risk of progression to end-stage liver disease and HCC can be heavily affected.

## 2. Definition

### 2.1. NAFLD

NAFLD refers to the excessive accumulation of triglycerides in the hepatocytes (hepatic steatosis) not secondary to alcohol damage in subjects who do not drink or do not have significant alcohol consumption [4,5]. Other causes of liver steatosis must be absent including B and C (genotype 3 in particular) viral hepatitis [6,7,8,9], hepatotoxic drugs such as amiodarone, anti-retroviral agents for HIV, glucocorticoids, methotrexate, tamoxifen, valproate [10], Wilson’s disease [11], parenteral nutrition [12], and starvation as a primary cause. 

Triglycerides (TG) can physiologically accumulate in the hepatocyte as lipid droplets, but liver steatosis points to the accumulation of TG exceeding the 95th percentile, i.e., >55 mg per g of liver tissue. Depending on the diagnostic methodology, liver steatosis occurs by histology with intracellular triglycerides ≥5% of hepatocytes [13,14], by magnetic resonance imaging proton density fat fraction (MRI-PDFF) when the estimated liver fat content is ≥5%, and by magnetic resonance spectroscopy when the estimated liver fat content is ≥5.56% [15].

Many epidemiological studies show that NAFLD has become one of the most common chronic liver disorders in Western countries [13,16,17,18], and accounts for 10 to 46% in the USA [16,17,18]. The medical costs related to NAFLD are high, about 103 billion USD in the United States and 35 billion USD in central Europe [19]. The median prevalence of NAFLD is about 25% worldwide [20,21,22], and figures are increasing with time [20]. This trend is likely due to the occurrence in North America and Europe of the increasing prevalence of major metabolic risk factors which include overweight, obesity with the expansion of visceral adiposity, type 2 diabetes mellitus, sedentary lifestyles, dyslipidemia, and metabolic syndrome [20,23,24,25]. NAFLD occurs in more than 90% of morbidly obese subjects [26] but “lean” NAFLD and NASH can occur in lean subjects as well [24] with a prevalence of 10–30% in both Western and Eastern countries [27,28]. 

There is an ongoing debate about the definition of “non-alcoholic” in NAFLD. The term overemphasizes “alcohol” rather than metabolic risk factors [4,29]. In this context, the term NAFLD describes a condition recently termed metabolic-dysfunction-associated fatty liver disease (MAFLD), where hepatic steatosis is associated with at least one of the following three comorbidities: overweight/obesity, presence of type 2 diabetes mellitus, evidence of metabolic dysregulation [22,30]. Research is looking at the contribution of environment, comorbidities, and the gut microbiome to the pathogenesis and natural history of NAFLD/MAFLD [31,32,33,34]. Additional factors contributing to NAFLD include the environment, gut microbiome, deranged glycolipid metabolic pathways, metabolic inflammation primarily mediated by innate immune signaling, comorbidities, and genetic risk factors [35,36]. 

The prevalence of NAFLD is underestimated, since many studies are based on the occurrence of mild hypertransaminasemia and/or on ultrasonographic steatosis [20], without exact quantitative evaluation of liver fat. Normal serum alanine aminotransferase (ALT) often occurs in NAFLD patients. Abdominal ultrasonography, on the other hand, has poor accuracy for mild steatosis (<30%) and necro-inflammatory changes typical of steatohepatitis, while it can easily detect a hyperechoic texture (“bright liver”) when a diffuse fatty infiltration is present [37]. 

NAFLD manifests with a large spectrum of abnormalities [38] ranging from simple steatosis named nonalcoholic fatty liver (NAFL) to nonalcoholic steatohepatitis (NASH), to liver cirrhosis which puts the patients at risk of hepatocellular carcinoma (HCC). NAFL occurs in about 80% of NAFLD subjects and is the non-progressive form of NAFLD since the risk of progression to liver cirrhosis is minimal [39]. There is little or no inflammation and no evidence of hepatocellular injury. NASH occurs less often, in about 20% of NAFLD subjects, and is characterized by steatosis, liver inflammation, and hepatocellular injury with ballooning and apoptosis. These features are indistinguishable from those of alcoholic steatohepatitis [40,41]. A feature of NASH is that patients are at high risk of developing liver fibrosis [42,43,44,45]. This possibility exposes patients to the risk of progression to cryptogenic compensated/decompensated cirrhosis as well as to HCC [46,47,48]. One feature of NAFLD is the exposure of the populations to the increased risk of liver-related mortality. The clinical outcomes in NAFLD are worse in the presence of liver fibrosis, with an average of 14 years vs. 7 years in NAFL and NAFLD, respectively. Time of progression is even shorter in “rapid progressors” NAFLD, a group of 10–20% of patients [39]. Hypertransaminasemia, morbid obesity, genetic susceptibility with a family history of cirrhosis in first-degree relatives, diabetes, and host microbiota are important predictor factors likely involved [49,50,51,52]. In addition, in cryptogenic cirrhosis, the yearly incident risk of developing HCC is 1.5–2.0%. Thus, NASH patients require careful screening [53], and this policy is supported by clinical data which show that NAFLD has become the second leading indication for liver transplantation in the US, and the number of cases of NASH-related HCC is increasing [21].

In addition, and similarly to obesity, NAFLD is a predisposing disease to all-cause mortality due to increased risk of cardiovascular disease and extrahepatic malignancies [54,55,56]. Lean NAFLD subjects are also at increased risk of metabolic dysfunction and increased cardiovascular risk [24,57]. 

### 2.2. ALD

ALD refers to liver damage caused by excessive alcohol consumption, which represents a global healthcare problem. As a risk factor, alcohol represents about 5.9% of all deaths globally and more than 25% of deaths in the age group 20–39 years [58]. For ALD, the relationship between alcohol use and disease is highly exponential. ALD has a lifetime prevalence of about 18% among adults in the US. Data from the National Health and Nutrition Examination Survey found that the prevalence of ALD among adults in the US was 4% [59]. The progressing and increasing profile of liver disease including ALD in Europe is another worrisome trend. This trend initially reported by Terris et al. [60] is confirmed when comparing the death rate for liver cirrhosis and alcohol consumption over time for ages >15 years, ranging from 20–40 deaths per 100,000 people and 5–15 liters consumed per capita, respectively, and trends in each country (i.e., alcohol consumption decreasing, increasing, stable over time) [3]. ALD is a spectrum of several liver disorders which, similarly to NAFLD, includes alcoholic hepatitis, alcohol-associated steatosis, alcohol-associated steatohepatitis, and alcohol-associated cirrhosis [61]. ALD is the most common cause of liver-related death in Western populations and is a leading cause for liver transplantation in both Europe and the US [1]. Mortality related to ALD in the United States averaged 5.5 per 100,000 persons in 2012 [62,63]. Major concerns about ALD include the worldwide rising trend [64,65] and the problem of alcohol consumption in late adolescence [66]. 

Since alcohol consumption is the primary risk factor for ALD, key aspects to be considered include age and drinking pattern. Daily drinking was associated with an increased risk of alcoholic cirrhosis in men. Recent alcohol consumption rather than early consumption increased the risk of alcoholic cirrhosis. Wine appears to be associated with a lower risk of alcoholic cirrhosis, compared to beer and liquor [67]. 

Studies in animal models support an etiologic role for cytokines in the liver injury of ALD. Patients with alcoholic hepatitis and/or cirrhosis show elevated serum concentration levels of TNF-alpha and TNF-alpha-inducible cytokines/chemokines, such as IL-6, -8, and -18, and levels correlated with markers of the acute phase response, liver function, and clinical outcome [68]. Hepatocyte nuclear factor-4 acts as the major transcriptional factor for the regulation of some genes involved in the lipid metabolism and oxidative process [69].

Although in clinical medicine NAFLD and ALD tend to be considered two different entities, NAFLD and ALD have several features in common which include genetic risk factors, few pathophysiological pathways, and histological features [70,71]. 

## 3. Liver Disease in Obesity

There is a close relationship between obesity and NAFLD, as well as other liver diseases [72,73,74,75,76]. The ongoing metabolic disorder emerging with obesity might also exacerbate NAFLD or liver damage due to ethanol and/or viruses [77]. 

Measures of body fat distribution could predict liver disease regardless of other metabolic risk factors. A first approach when assessing patients with overweight/obesity is to measure the body mass index (BMI) which is easy to measure and correlates with the percentage of body fat and body fat mass [78], better than body weight alone [79]. The National Health Institute (NIH) and World Health Organization (WHO) have adopted BMI cut-off values in white, Hispanic, and black individuals for underweight (BMI < 18.5 kg/m^2^), normal weight (BMI 18.5 to 24.9 kg/m^2^), overweight (BMI 25.0 to 29.9 kg/m^2^), and obesity (>30.0 kg/m^2^). In the Asian population, values are lower to define overweight (BMI 23.0 to 24.9 kg/m^2^) and obesity (BMI > 25.0 kg/m^2^) [80,81,82]. Waist circumference and waist-hip ratio (WHR), at variance of BMI, are a more specific marker of abdominal (visceral) adiposity. Finding a waist circumference of ≥102 cm for men and ≥88 cm for women is considered elevated and indicative of increased cardiometabolic risk [83]. In addition, both waist circumference and WHR provide independent risk information that is not accounted for by BMI [84], e.g., increased risk for NAFLD but also heart disease, diabetes, hypertension, and dyslipidemia [85,86,87,88,89,90,91,92,93,94] and increased mortality rates [84,90,91,92,93,94]. 

It is clear that the liver is an important target of disease in obese individuals and that the onset and progression of damage are driven by metabolic factors, where hepatic steatosis is associated with at least one of the following three comorbidities: overweight/obesity, presence of type 2 diabetes mellitus, evidence of metabolic dysregulation [30,95,96]. In addition, the progression of NAFLD is strongly associated with the increasing number of metabolic syndrome (MetS) components [97], whereas a high BMI per se does not predict a more advanced NAFLD stage [39,98]. In this respect, WHR could be the strongest predictor of incident severe liver disease [92], since the effect of WHR appears to be independent of BMI and BMI brings no extra predictive value when compared with WHR [91,92,99]. The poor predictive value of BMI vs. liver diseases might be influenced by a U-shaped, rather than linear association with liver disease [75]. By contrast, “pear-shaped” obesity reflects metabolically healthy obesity and less visceral fat inflammatory changes and less liver fibrosis than “apple-shaped” (metabolically unhealthy) obesity [96,100].

Notably, the best predictors of incident advanced liver disease in non-risk alcohol drinkers were abdominal obesity with HR 1.03 per cm increase in waist circumference, insulin resistance with HR 1.04 per unit increase in homeostasis model of assessment of insulin resistance, and a low LDL (low-density lipoprotein) cholesterol level with HR 0.54 per mmol/L increase in LDL cholesterol) [90]. Lastly, the features of the MetS also predict HCC and survival in CLD irrespective of the primary etiology of the liver disease [101,102].

A recent important case-control study based on 1293 patients with alcohol-associated cirrhosis and 754 individuals with similar lifetime alcohol exposure but without liver disease, the authors found that both metabolic and lifestyle factors were linked to the risk of alcohol-associated cirrhosis [103]. Cirrhotics were more likely to have diabetes (odds ratio (OR) 3.09, 95% CI 2.02–4.72) and higher premorbid body mass index (OR 1.06, 95% CI 1.03–1.09). Cirrhotic patients were less likely to drink coffee or tea. The results suggest therefore that lifestyle interventions such as weight loss might become valuable tools to reduce the risk of alcohol-associated cirrhosis.

## 4. Alcohol Metabolism

The liver is the main site for alcohol metabolism which involves mainly two enzymes: alcohol dehydrogenase (ADH) and aldehyde dehydrogenase (ALDH). First, ADH metabolizes alcohol to acetaldehyde, a highly toxic substance and known carcinogen. Then, in a second step, acetaldehyde is further metabolized to a less active by-product called acetate. Other than in the liver, ADH is also detected in the stomach where it contributes to the first step of ethanol metabolism. Ethanol metabolism includes both oxidative and non-oxidative pathways. The oxidative metabolism of alcohol in the liver (Figure 1) involves three different pathways and cell compartments:
(1)Cytosol: the enzyme ADH is responsible for most of the ethanol metabolism. The reaction involves an intermediate carrier of electrons, nicotinamide adenine dinucleotide (NAD+), which is reduced by two electrons to form NADH.(2)Endoplasmic reticulum: the enzyme cytochrome P450 2E1 (CYP2E1) is NADPH cofactor-dependent and belongs to the microsomal ethanol oxidizing system (MEOS). However, CYP2E1 is only active after a person has consumed large amounts of alcohol. This pathway is important in metabolizing ethanol to acetaldehyde at elevated ethanol concentrations.(3)Peroxisomes: the enzyme catalase uses hydrogen peroxide (H_2_O_2_) to oxidize alcohol. Catalase metabolizes only a small fraction of alcohol in the body.

Acetaldehyde is a by-product of ethanol metabolism which is a highly toxic and carcinogenic compound that forms adducts with DNA (e.g., 1,N2-propanodeoxyguanosine), lipids, and proteins and involves microsomal proteins and microtubules. Acetaldehyde also forms adducts with the brain signaling chemical (i.e., neurotransmitter) dopamine to form salsolinol, (which may contribute to alcohol dependence). Acetaldehyde is subsequently detoxified to acetate by aldehyde dehydrogenase 2 (ALDH2) in mitochondria to form acetate and NADH. Acetate is oxidized to carbon dioxide (CO_2_) but most acetate escapes the liver to the blood and is eventually metabolized to CO_2_ in the heart, skeletal muscle, and brain cells. Acetate increases blood flow into the liver and depresses the central nervous system, as well as affecting various metabolic processes [106]. Acetate, also metabolized to acetyl CoA, is involved in lipid and cholesterol biosynthesis in the mitochondria of peripheral and brain tissues.

The non-oxidative pathway of ethanol metabolism is catalyzed by the enzyme fatty acid ethyl ester synthase (FAEES) with the formation of fatty acid ethyl ester (FAEE). The second pathway involves the enzyme phospholipase D to form phosphatidyl ethanol. The role of such pathways in tissue damage remains to be fully elucidated, but inhibition of ethanol oxidation acting on ADH, CYP2E1, and catalase results in an increase in the nonoxidative metabolism of alcohol and increased production of FAEEs in the liver and pancreas [107].

## 5. Mechanisms of Damage

The detrimental effect of alcohol metabolism in the liver includes hypoxia, generation of harmful adducts, highly reactive oxygen species (ROS) which can damage other cell components, and changes in the NADH/NAD^+^ ratio which is a marker of cellular redox state [108].

Upon chronic alcohol intake, the brain starts using acetate rather than glucose as a source of energy.

### 5.1. Steatosis

As for NAFLD, alcohol-associated steatosis implies decreased fatty acid oxidation and increased lipogenesis [109], impaired beta-oxidation, and tricarboxylic acid cycle activity [110]. The chronic use of ethanol will increase key lipogenic enzyme regulators (sterol regulatory element-binding proteins (SREBPs)), as well as SREBP-1 target genes, such as fatty acid synthase, acetyl-coenzyme A carboxylase, and stearoyl-CoA desaturase [111]. The ethanol-induced lipogenic/steatogenic pathway might be complicated by the underlying metabolic-dependent steatosis (see below).

### 5.2. Steatohepatitis

The evolution to steatohepatitis involves several steps such as the innate immune system [112], with the deranged function of dendritic cells, release of proinflammatory cytokines [113], and abnormal function and response of neutrophils, macrophages (phagocytosis, apoptotic regulation, and arachidonic acid metabolism [114]), natural killer cells, natural killer T cells, and T lymphocytes [115].

Further elements involved include hypoxia, formation of adducts, oxidative stress, endotoxins, production of cytokines, COX-2 enzymes microbiota, dietary fat, microRNAs, and impaired liver regeneration.

Notably, NASH patients show an increased abundance of alcohol-producing bacteria in microbiomes (Proteobacteria, Enterobacteriaceae, and Escherichia), and findings are associated with significantly elevated blood ethanol levels [116,117]. Gut dysbiosis might also explain the finding of small intestinal overgrowth and endotoxemia and its relation with TLR signaling genes and liver histology in patients with NAFLD [118].

Hypoxia is a predisposing factor to alcohol damage in the liver. Alcohol metabolism relies mainly on ADH and ALDH. The generation of NADH in the mitochondria requires oxidation with activation of the mitochondrial electron transport system or respiratory chain, and transfer of electrons to molecular oxygen (O_2_), which binds H+ to generate H_2_O. Ethanol metabolism tends to increase blood oxygen uptake by the hepatocyte [119], but this step might be associated with hypoxia in the perivenous hepatocytes [120,121]. In addition, ethanol can activate Kupffer cells in the liver and indirectly increase O_2_ consumption via release of prostaglandin E2 and increased hepatocyte metabolic activity of hepatocytes which also require oxygen.

The role of the innate immune system and cellular involvement is clear in ALD. Neutrophil infiltration is an important characteristic in alcoholic hepatitis and an increase in the peripheral neutrophil count can occur as well [122]. At least two chemoattractants are involved. Interleukin-8 (IL-8) is elevated in plasma in patients with alcoholic hepatitis [122,123,124] and correlates with the severity of hepatic injury [122]. Another chemoattractant of neutrophil is an arachidonic acid metabolite related to leukotriene B4 [122], released by cultured hepatocytes challenged with ethanol [125]. The sequence of release might involve the production of ROS in MEOS with an attack of free radicals on membranes [126].

Adduct formation plays an important role in the evolution towards alcoholic steatohepatitis. In chronic alcohol users, the ADH/ALDH pathway becomes saturated. Under this condition, reactive aldehydes are produced from the metabolism process, e.g., malondialdehyde-acetaldehyde (MAA), 4-hydroxy-2-nonenal (HNE), and lipid hydroperoxides which can bind to proteins to produce protein adducts [105]. Metabolism of ethanol by ADH and CYP2E1 produce reactive molecules (e.g., acetaldehyde and ROS). Ethanol metabolites and adducts generated during ethanol metabolism include therefore acetaldehyde, malondialdehyde (MDA) generated by nonenzymatic lipid peroxidation of unsaturated fatty acids, breakdown of arachidonic acid in platelets, 4-hydroxynonenal (HNE) generated by lipid peroxidation of long-chain polyunsaturated fatty acids, malondialdehyde-acetaldehyde adduct (MAA) generated by hybrid adducts with malondialdehyde and acetaldehyde, and hydroxyethyl radical (HER), generated by ethanol oxidation in the presence of iron. The antibodies to acetaldehyde–lysine adducts bind to the adducts and induce the immune system to destroy the hepatocytes containing these adducts. This process is known as immune-mediated hepatotoxicity or antibody-dependent cell-mediated cytotoxicity (ADCC). Antibodies directed against other acetaldehyde–protein adducts also have been found in the blood of alcoholics [127,128]. MDA, HNE, MAA can induce immune responses and inflammatory in stellate cells and endothelial cells and be mediators of liver damage [129,130].

The formation of ROS is also associated with mitochondrial damage and further activation of Kupffer cells with the release of TNF-a, a mediator of liver fibrosis. Mitochondrial damage (increased permeability) releases cytochrome c into the cytosol which mediates apoptosis and, via decreased ATP, leads to necrosis.

CYP2E1-mediated ethanol metabolism generates oxidative stress leading to DNA damage, a pathway to alcohol-related development of liver cancer [131].

Chronic ethanol consumption and alcohol metabolism contribute to impairment of metabolic processes and metabolic disorders such as hyperlipidemia, lactic acidosis, increased levels of ketones (ketosis), and hyperuricemia. Within the spectrum of ALD, and similarly to NAFLD, liver steatosis may evolve inflammation, apoptosis, fibrosis, and eventually cirrhosis. Fatty liver is due to the shift in the redox state of the hepatocytes that results from ethanol metabolism by ADH, with an accumulation of fatty acids, rather than their oxidation. Ethanol also changes the function of proteins that help regulate fatty acid synthesis and oxidation [132].

Ethanol-induced activation of ADH and ALDH is associated with inhibition of retinol metabolism which is important to the normal structure and function of stellate cells in the liver [133].

Chronic alcohol increases the risk of liver cancer [134] and mechanisms might include increased induction of acetaldehyde, CYP2E1, and ROS formation [135].

Adduct formation might result in dysfunctional intracellular proteins [136] and acetaldehyde protein adducts may act as neoantigens which induce both cell-mediated and humoral immune responses to attack cells bearing these compounds [137]. Notably, acetaldehyde-protein adducts are also detected in patients with nonalcoholic liver disease.

Hydroxyethyl radical adducts are also involved in alcohol-induced liver injury [138]. Another toxic metabolite of lipid peroxidation is acrolein when not cleared through conjugation to glutathione. Acrolein is responsible for endoplasmic reticulum stress that leads to cell damage and death [139].

Oxidative stress depends on cellular damage due to ROS and plays a role in alcohol-induced liver injury and fibrosis [140,141,142]. Sources of ROS are the mitochondria and CYP2E1 (hepatocytes) and nicotinamide adenine dinucleotide phosphate (NADPH) oxidase in activated Kupffer cells [110].

Chronic alcohol consumption leads to the induction of microsomes (intracellular membrane compartments, such as smooth endoplasmic reticulum) due to spillover from the ADH pathway. Induction of cytochrome CYP2E1 within the microsomal ethanol oxidation system (MEOS) [143] occurs in individuals with alcoholic liver disease [144] and contributes to the generation of oxidative stress [145,146]. Both ADH and CYP2E1 are more concentrated in the centrizonal region of the hepatic lobule, the area in which alcohol-induced liver injury is most prominent. Increased inducible nitric oxide synthase (iNOS) levels occur during chronic ethanol ingestion and leads to nitric oxide production [147] which reacts with reactive oxygen species in the liver [110]. The epigenetic modulator, sirtuin 6 (SIRT6), might protect the liver from oxidant stress during alcohol-induced liver injury [148]. The molecule metal regulatory transcription factor (Mtf1) enhances SIRT6 activity.

SIRT1 decreases in the livers of aging mice and the liver becomes more sensitive to oxidative stress and injury as well as amplified fibrosis [149].

Endotoxins originate from the gut microbiota, which contributes to alcohol-associated liver disease [150,151,152] and one piece of evidence is the increased serum level of endotoxin (lipopolysaccharide (LPS)) in alcoholic patients with liver disease and animal models exposed to alcohol [153]. The sequence involves Toll-like receptors, endoplasmic reticulum stress, and the inflammasome [154,155,156], with endotoxin interacting with LPS-binding protein and receptor CD14 on the surface of liver macrophages (Kupffer cells). CD14 then interacting with toll-like receptor type 4 (TLR4), leading to cytokine activation. Animal models support these mechanisms [157,158,159]. Toll-like receptor 3 (TLR3) may attenuate alcohol-induced liver injury [160]. Other animal studies have challenged this hypothesis [161].

Increased levels of the proinflammatory cytokines tumor necrosis factor-alpha (TNFa), interleukin-1, and interleukin-6 [122,162] might influence the hemodynamic alterations of patients with cirrhosis. Injury and fibrosis might involve the p90RSK kinase pathway [163]. TNFa might be one mediator of alcohol-induced liver injury [164,165]. TRAIL (tumor necrosis factor-related apoptosis-inducing ligand) may also contribute to the pathogenesis of steatosis and liver injury [166]. Interleukin-17A (IL-17) is another pro-inflammatory cytokine that activates macrophages and regulates steatosis in the liver [167].

The enzyme cyclooxygenase 2 (COX-2) produces prostaglandins and is elevated in experimental alcohol-induced liver injury [168]. Ethanol-fed rats have increased expression of cyclooxygenase 2 mRNA in Kupffer cells, with the production of thromboxanes [169,170]. Indeed, thromboxane inhibitors attenuate some of the pathologic changes in these animals [169].

Dietary, saturated fat is capable of down-regulating liver injury in experimental alcohol feeding, likely due to reduced endotoxemia and lipid peroxidation, which in turn result in decreased levels of TNF-alpha and Cox-2 [168]. Fat metabolism might play a role in the pathogenesis of ALD since the *PNPLA3* gene (adiponutrin) increases the risk of both nonalcoholic [171] and alcohol-associated liver disease [172].

MicroRNAs are non-coding RNA species acting through epigenetic mechanisms. MicroRNA-182 is the most highly expressed miRNA in alcoholic hepatitis. MicroRNA-182 expression is associated with disease severity, short-term mortality, and the ductular reaction of alcoholic hepatitis [173]. Conversely, the microRNA-122 is downregulated in patients with alcoholic hepatitis and is also involved in the progression of the disease [174].

Impaired liver regeneration might be also involved in ALD. Upon chronic ethanol ingestion, the liver slowly loses the capacity to regenerate. The hepatocyte transcription factor HNF4-alpha plays a role in liver regeneration [174] and livers of patients with progressive stages of alcohol-associated liver disease show loss of activity of downstream HNF4-alpha gene targets through DNA hypermethylation, which is an epigenetic modification [174]. Dysfunctional HNF4-alpha might follow the increased activity of the fibrogenic cytokine transforming growth factor beta 1 (TGFb1). Indeed, forced expression of HNF4-alpha was found in the liver of rescued rats with severe fibrosis and irreversible liver failure [175].

Gut dysbiosis and bacterial products influence the liver before thorough drainage of gut contents into the portal vein. Advanced liver disease is associated with increased permeability with translocation of intestinal products to the liver. In animal models, the manipulation of the fecal microbiome in murine alcohol-induced liver injury can alter the extent of injury [176]. In humans, the imbalance of different bacterial species has been demonstrated [151]. Candida overgrowth is a condition leading to increased production of pathogenic endotoxin [177,178]. *Enterococcus faecalis* might be involved in hepatocyte death and liver injury through the production of a cytolysin [179]. More studies should characterize the microbiome characteristics in ALD.

## 6. Fibrogenesis

The cellular sources of the extracellular matrix (ECM) pave the way to fibrosis in alcohol-induced liver injury. Different injuries involve alcohol, drugs, viral infection, or metabolic abnormalities (such as hemochromatosis). Involved are “interstitial” collagens (types I and III), glycoproteins (laminin and fibronectin), and proteoglycans (dermatan and chondroitin sulfate).

The hepatic stellate cell is the principal source of the ECM in hepatic fibrosis, including ALD [180,181,182]. In health, stellate cells are perisinusoidal cells distributed throughout the liver serving to store hepatic vitamin A. Activated stellate cells during injury lose vitamin A, proliferate, and become fibrogenic [181,183,184].

The release of cytokines from neighboring cells (paracrine activation) and alterations in the surrounding cellular microenvironment contribute to activation [181], and major cytokines involved are platelet-derived growth factor (PDGF), transforming growth factor-1, and endothelin-1.

Acetaldehyde has minor fibrogenic activity towards cultured stellate cells [185]. Lipid aldehydes are unstable products between reactive oxygen intermediates and cellular proteins. 4-hydroxynonenal and malondialdehyde increase with alcohol feeding in rodents [186,187,188,189] and may have fibrogenic activity.

## 7. Multiple Effects of Alcohol

Since many individuals whose alcohol consumption exceeds the thresholds do not develop liver disease, other factors must operate and be associated with increased risk of alcohol-associated liver injury [190], and include, for example, overweight, obesity, underlying NAFLD, and tobacco use [191]. As stated previously, a common and potential pathway of histological progression for both NAFLD and ALD is steatosis, steatohepatitis, fibrosis, cirrhosis, and even HCC. A striking aspect of NAFLD, however, is the apparent discrepancy between the overall high prevalence of disease (especially in obese subjects) and the small number of NAFLD patients evolving towards complicated liver disease, i.e., less than 5% [192]. On the other hand, while the mortality from liver cirrhosis is greatly associated with per capita alcohol consumption in the population [193], only 10 to 15% of heavy drinkers develop cirrhosis during their lifetime [194,195].

Both obesity and alcohol consumption might represent intersecting aspects between NAFLD and ALD which can increase the chance to develop complicated liver disease. In this regard, the prevalence of both obesity and overweight have nearly tripled since the 1970s worldwide [1], and the average prevalence of obesity in OECD (Organization for Economic Cooperation and Development) countries is 23%. However, in the US, the prevalence rate is the highest, i.e., 38% [196]. Another aspect is alcohol consumption with around 60% of adults in Europe and the United States drinking alcohol, and per capita annual consumption among the active drinkers (abstainers excluded) averages 15 to 17 L of pure alcohol [64].

The consequence of the above-mentioned aspects is that several subjects with metabolic disorders and obesity also drink alcohol and this combination will pave the way to the detrimental and synergistic effect of alcohol and obesity and to the development of liver disease [90,197,198,199]. In this scenario, metabolic risk might sensitize the liver to alcohol damage and vice versa [90,197,198,200].

### 7.1. Alcohol and Metabolic Disorders

In addition to the intrinsic harmful effect of ethanol, the caloric impact of this agent cannot be neglected. One gram of ethanol equals 7 kcal, while a standard drink is any drink (ranging from ~5% of beers to ~40% of spirits) that contains 14 g of pure alcohol [201]. Thus, one standard drink would contain about 98 kcal and, depending on the number of daily drinks, ethanol can significantly contribute to excess of dietary calories. Whereas light-to-moderate alcohol intake does not necessarily lead to weight gain, heavy drinking is associated with significant weight gain [202,203]. Thus, a daily intake of >500 mL of beer appears to be associated with abdominal obesity, namely “beer belly” [204]. A feature in subjects with chronic alcohol abuse or cirrhosis is the reduced overall fat mass due to lipid oxidation to meet energy requirements due to increased energy expenditure [205]. The role of lifestyle, with respect to alcohol consumption and body weight, should not be underestimated, since alcohol could deeply influence aspects related to psychological aspects, regulation of food intake, psychosocial well-being, sleeping, and depression symptoms, with substantial interindividual variation [202,203].

### 7.2. ALD and Obesity

By an arbitrary definition, a daily alcohol intake below the threshold of 30 g/day for men and 20 g/day for women seems to be insufficient to induce hepatic steatosis, and this cut-off discriminates between ALD and NAFLD [4,13,206]. Indeed, the current definition of NAFLD rules out “heavy” alcohol consumption, B and C viruses, several drugs, Wilson’s disease, and starvation as a primary cause. By these amounts of alcohol, the risk is equally low for alcoholic cirrhosis [193].

A first aspect to consider is the methodological aspect due to a pre-selection bias of ALD groups/cohorts where patients with “pure” alcoholic liver cirrhosis [207,208,209] have a history of consuming 6 to 12 drinks daily for an average of 30 to 40 years. In this scenario, patients drinking much lower amounts of alcohol and with additional metabolic disturbances affecting the liver could be easily excluded from the studies.

In addition, the pattern of alcohol consumption, however, can dramatically vary across populations, cultures, lifestyles, location, gender, and other demographic factors.

In this respect, drinking patterns can influence alcohol toxicity with factors which include quantity and frequency, beverage type (with lower risk for wine), drinking while fasting, and binge drinking [67,193,210,211,212].

Such factors might not be the only ones since individuals with longstanding alcohol-use disorder but without liver disease have drinking patterns comparable to patients with ALD [207,208].

Additional factors involved in alcohol-induced liver damage could therefore include gut dysbiosis in both ALD and NAFLD [213], as well as genetics, gender, environmental factors, and diet. There is accumulating evidence suggesting that several underlying metabolic abnormalities as obesity and MetS must be involved as well in ALD progression [199]. For example, independently of alcohol consumption, risk drinkers from the general population without manifest liver disease developed advanced liver disease during follow-up. This evolution occurred independently from alcohol consumption and damage was predicted by classic metabolic abnormalities, namely abdominal obesity (HR: 1.6 per 1 SD increase in WHR), diabetes (HR: 3.4), hypertension (HR: 2.5), dyslipidemia (HR: 2.1 for HDL cholesterol; HR: 1.2 for triglycerides), and sedentary behavior (HR: 1.6 for exercise less than once monthly) [214]. In addition, distinct metabolic abnormalities can decrease or disappear with ongoing end-stage liver disease, which include obesity, dyslipidemia, and hypertension. This possibility might affect the interpretation of several studies dealing with ALD and metabolic NAFLD. In line with such possibility, studies have shown that the rate of NAFLD recurrence after liver transplant was higher in ALD patients on abstinence, as compared to patients without ALD (37 and 26%, respectively). Findings suggest the importance of underlying metabolic stressors/risk factors in ALD patients, and therefore confirm the ongoing synergistic effect of both conditions (alcohol and metabolic disorders, changes of gut microbiota, etc.) [215,216].

Indeed, ALD and NAFLD share pathogenetic similarities at a molecular level [217] and undistinguishable histologic findings, on a background of multiple mechanisms contributing to disease progression [218,219]. Mitochondrial dysfunction and oxidative stress are additional mechanisms of liver damage and link the progression from simple steatosis to the necro-inflammatory form of steatohepatitis [220]. There are some similarities in the gut microbiota composition in patients with NAFLD and ALD, such as dysbiosis, bacterial overgrowth, altered intestinal permeability, Farnesoid X receptor (FXR) signaling, increased primary and secondary bile acids, decreased phosphatidylcholine, increased *Enterobacteriaceae* and decreased *Akkermansia muciniphila* [221], and endotoxemia [32,219,222,223]. The genetic background could share similarities in ALD and NAFLD [70].

Adiponutrin (*PNPLA3*) is a gene regulated by energy balance in human adipose tissue [224], and a common variant in *PNPLA3* (rs738409[G]), which encodes adiponutrin, is associated with liver fat content in humans [225]. Common polymorphism in the *PNPLA3*/adiponutrin gene confers higher risk of cirrhosis and liver damage in ALD [225,226] and NAFLD as well [36,51,225,227]. The rate of progression of fatty liver disease to fibrosis and cirrhosis and HCC is approximately two- to threefold among carriers of the risk variant both in NAFLD and ALD [70,228,229,230,231]. Another genetic risk variant (rs72613567:TA in HSD17B13 encoding the hepatic lipid droplet protein hydroxysteroid 17-beta dehydrogenase 13) could also be involved in NAFLD and ALD [70,232] with reduced levels of ALT and AST, reduced risk of ALD (by 42% among heterozygotes and by 53% among homozygotes), NAFLD (by 17% among heterozygotes and by 30% among homozygotes), alcoholic cirrhosis (by 42% among heterozygotes and by 73% among homozygotes), and nonalcoholic cirrhosis (by 26% among heterozygotes and by 49% among homozygotes) [232]. Other genetic variants are also involved, such as *TM6SF2* (associated with hepatic steatosis and cirrhosis in patients with NAFLD and ALD), the membrane bound O-acyltransferase domain-containing 7 (*MBOAT7*) (progression of NAFLD and ALD), and other variants related to the genes involved in insulin resistance, lipid metabolism, glucose metabolism, oxidative stress, inflammatory pathways, and fibrosis playing as disease modifiers in patients with NAFLD and ALD. Epigenetic factors could involve microRNAs and DNA methylation acting on disease course in NAFLD and ALD [225].

The most important evidence linking ALD to NAFLD is summarized in Table 1.

### 7.3. Alcohol and NAFLD

Data from clinical studies and population habits have provided conflicting results about the possible link between alcohol consumption vs. obesity and vs. liver diseases. Either deterioration of steatosis [200,242] and hypertransaminasemia [243,244,245] or reduced NAFLD prevalence and liver fibrosis upon light consumption of ethanol [246,247,248,249,250] have been described. A potential explanation for the putative protective effect in the latter case would imply that light alcohol intake is associated with improved insulin sensitivity, better lipid profiles, and anti-inflammatory effects [251,252]. Ajmera et al. [253] reviewed this topic and underscored a number of important limitations (Table 2).

## 8. Genetic Predisposition

Genetic predisposition is one of the modifiers of the progression of ALD. In 2010, Tian et al. genotyped 305 individuals with alcohol dependance albeit normal function of the liver (controls), 482 individuals with ALD-cirrhosis, and 434 with ALD but without cirrhosis [172]. Analysis of the common genetic variant p.I148M in the *PNPLA3* (adiponutrin) gene, previously detected risk factor for NAFLD [171], was significantly associated with ALD and ALD-cirrhosis. This association was then replicated with several subsequent studies which were analyzed in a meta-analysis by Salameh et al. [254], demonstrating that carriers of the *PNPLA3* risk allele have higher odds of alcohol-induced liver injury, alcoholic cirrhosis, and hepatocellular carcinoma (HCC). Moreover, other variants that are known to modulate the risk of fatty liver [255] were analyzed in patients with alcoholic liver disease. For example, in a study encompassing 325 individuals with history of excessive drinking the *TM6SF2* p.E167K polymorphism was associated with increased liver fibrosis in a multivariate model including patient’s age, HOMA-IR, LDL cholesterol, and *PNPLA3* p.I148M [256]. TM6SF2 is localized in enterocytes and hepatocytes which synthesize apolipoprotein B-containing lipoproteins. The TM6SF2 p.E167K variant is on one side associated with increased hepatic fat and on the other side with lower serum LDL cholesterol and triglycerides [257,258]. Other variants, such as *GCKR* p. P446L and *MBOAT7* rs641738, have also been shown to enhance the risk of developing fatty liver [259]. MBOAT7 is a lysophosphatidylinositol acyltransferase activity and is linked to levels of arachidonic acid [260]. Interestingly, not only harmful genetic variants in the setting of ALD were detected. Abdul-Husn demonstrated in 2018 [232] the *HSD17B13* truncating variant might be protective against fatty liver and alcoholic cirrhosis. The same variant was also shown to reduce the risk of HCC in the setting of ALD [261]. In addition to *HSD17B13*, the *MARC1* gene emerged as a new potential protective gene in fatty liver. This was also shown in patients with ALD: analysis of data from the UK Biobank demonstrated that minor MARC1 rs26424438 decreases the odds of developing liver cirrhosis in patients with ALD [262]. The function of MARC1 (mitochondrial amidoxime reducing component 1) is not fully understood but carriers of the protective minor allele seem to be characterized by higher hepatic polyunsaturated phosphatidylcholines [263]. Hence, according to the currently available data in the literature, there are several genetic variants with harmful effects in patients with ALD, whereas other variants seem to have protective effects. Hence, polygenic scores are required to stratify patients’ risk of ALD in relation to the inherited predisposition. Such scores can also encompass already available non-invasive fibrosis and steatosis scores [264]. Nevertheless, genetic analyses have not gained much attention and are not commonly used in the clinical work-up of patients with ALD and NAFLD [255].

## 9. Extrahepatic Manifestations

Two prospective cohort studies in the USA reported that light to moderate drinking is associated with a minimally increased risk of overall cancer. In particular, for men who have never smoked, the risk of alcohol-related cancers is not appreciably increased for light and moderate drinking (up to two drinks per day) but for women who have never smoked, the risk of alcohol-related cancers (mainly breast cancer) increases even within the range of up to one alcoholic drink a day [265]. Low alcohol intake (1–2 drinks per day for women and 2–4 drinks per day for men), however, is inversely associated with total mortality in both men and women, in terms of survival [266]. Another study on the NAFLD cohort confirmed both of these associations [267]. When interpreting the results of such studies, however, the role of confounding factors should be considered and another study in NAFLD patients found no association between alcohol use and the presence of cardiovascular risk factors or subclinical cardiovascular disease [268], although this study did not analyze clinical cardiovascular events.

A recent analysis of 599,912 current drinkers combined analysis of participant data from three large-scale data sources in 19 high-income countries (the Emerging Risk Factors Collaboration, EPIC-CVD, and the UK Biobank). The results were derived from 83 prospective studies assessing risk thresholds for alcohol consumption [269]. The authors found that for current drinkers of alcohol in high-income countries, the threshold for the lowest risk of all-cause mortality was about 100 g/week. However, for cardiovascular disease subtypes other than myocardial infarction, the alcohol intake level with the lowest risk was zero. According to this study, limits of alcohol consumption are lower than those recommended in most current guidelines.

In NAFLD patients, data are more conflicting on the association between low alcohol intake and all-cause mortality [212,270,271,272].

A Finnish study found a J-shaped association [270] and even low alcohol intake in fatty liver disease was associated with increased risks for advanced liver disease and cancer. Low to moderate alcohol use was associated with reduced mortality and CVD risk but only among never smokers.

Two studies from the US gave contradictory results. Younossi et al. [271] collected data from the National Health and Nutrition and Examination Survey III and could not show any mortality among light drinkers. In contrast, the association of alcohol consumption with increased mortality in participants with fatty liver (liver steatosis by ultrasonography) and metabolic syndrome points to an overlap between non-alcoholic and alcohol-related fatty liver disease.

Another study also based on the National Health and Nutrition Examination Survey data [272] described a J-shaped association between drinking and all-cause mortality. Among patients with NAFLD (assessed by hepatic steatotic index), modest alcohol consumption was associated with a significant decrease in all-cause mortality, whereas drinking ≥1.5 drinks per day was associated with increased mortality. These results help to inform the discussion of potential risks and benefits of alcohol use in patients with NAFLD.

In patients with NAFLD cirrhosis, low use of alcohol (<30 g/day for men and <20 g/day for women) compared with abstinence was associated with an increased risk of death or liver transplantation (HR 2.3), hepatic decompensation (HR 1.7), and HCC (HR 3.2). This study supports the strategy for absolute alcohol abstinence in patients with liver cirrhosis. [273].

## 10. Synergistic Effects of ALD and NAFLD

In clinical medicine, the additive role of metabolic disorders, including obesity, alcohol intake, and liver disease cannot be underestimated.

There must be a continuum between categories of NAFLD and ALD, with overlap and interactions as shown by a large study from the Finnish general population who developed the incident advanced chronic liver disease during 10 years of follow-up [274]. In about 39% of the overall cohort of 38,801 subjects with 174 subjects who developed incident advanced liver disease, the effect of ethanol and metabolic abnormalities could not be dissected but rather depended on joint effects of alcohol and metabolic risk.

As mentioned before, moderate drinking leads to increasing harm to the liver with metabolic risk factors [90]. In this context, a simple dichotomous approach to either non-alcoholic or alcoholic liver disease is not going to recapitulate the importance of combined factors in the genesis of liver disease progression [214].

Strategies should aim for a better assessment of individual risk for developing clinical liver disease and assessment of response to therapeutic interventions. Areas of research should include the design of large longitudinal studies, with alcohol consumption assessed at different time points and by more objective biomarkers such as phosphatidylethanol. Evaluation should include also factors influencing individual susceptibility to alcoholic liver toxicity, the role of genetics, chronic alcohol consumption, binge drinking, and metabolic abnormalities on liver toxicity in NAFLD patients. Risk stratification concerning combined and synergistic effect of metabolic factors and alcohol intake represent additional aspects to be investigated.

To date, the effects of alcohol are detrimental and extend beyond simple steatosis. Low alcohol consumption does not protect from liver disease, while the type of damage is influenced by aspects connected with genetics, intake pattern, beverage type, lifetime consumption, and comorbidity. Further implications are that low alcohol intake is associated with liver cancer risk. Thus, total alcohol abstinence is advisable in advanced liver fibrosis or subgroups at risk for progressive liver disease. The role of metabolic and dietary factors in the progression of liver disease must be addressed also among active drinkers even when alcohol withdrawal is started [275].

NAFLD is the primary consideration in patients who deny alcohol use disorder but have clinical features suggestive of alcohol-associated liver disease (such as elevated aminotransferases in the absence of serum markers of viral hepatitis).

Especially if subjects deny alcohol use, it is difficult to differentiate between NAFLD and ALD, based on clinical or histologic features [276]. Macrovesicular steatosis (i.e., membrane-bound large droplet) and inflammation with Mallory–Denk bodies are similar between NAFLD and ALD [277,278]. Findings such as canalicular cholestasis, marked ductular reaction, acute inflammation in the portal regions, periportal fibrosis, and less steatosis or nuclear vacuolization are more common in ALD than NAFLD [279]. Early changes in patients with ALD seen under the electron microscope include accumulation of membrane-bound large droplet (macrovesicular) steatosis made of neutral triglycerides. Additional findings in ALD are a proliferation of smooth endoplasmic reticulum and gradual distortion of mitochondria [280] in the pericentral region of hepatic lobules (zone 3) [281]. At an early stage, fat accumulation is detected with light microscopy whereas inflammatory changes are minimal, apart from occasional lipogranulomata in pericentral zones [282]. If present, smaller lipid droplets may resemble microvesicular steatosis seen in other conditions, such as pregnancy, tetracycline toxicity, and Reye’s syndrome, and made of free fatty acids which are not seen on standard light microscopy, rather than neutral triglycerides. There is the possibility that steatosis may progress to steatohepatitis, where inflammation appears early in zone 3, and then to portal tracts. Neutrophils are a component of alcohol-associated steatohepatitis. Mallory–Denk bodies appear as eosinophilic accumulations of intracellular protein aggregates within the cytoplasm of hepatocytes. These bodies are made of condensations of intracellular “intermediate filaments” or cytokeratins that are normal components of the hepatocyte cytoskeleton [283] and also appear in nonalcoholic steatohepatitis (NASH) [277]. Mallory–Denk bodies become an important marker of alcohol-induced injury. In a previous study, the authors detected Mallory–Denk bodies in 76% and 95% of alcoholic hepatitis patients and cirrhotic patients, respectively [284].

Fibrosis in ALD first appears in zone 3 (hyaline necrosis) and may become panlobular if subjects continue to drink [194] and predicts a high likelihood of cirrhosis [285,286]. Cirrhosis appears as regenerative be micronodular or macronodular nodules [287,288] and is generally thought to be irreversible. Some patients with micronodular cirrhosis will later progress to macronodular cirrhosis [289].

Obesity per se is insufficient to differentiate ALD from NAFLD since the two conditions often coexist. The ALD/NAFLD index (ANI), a specific algorithm generated by logistic regression models, is a predictive model to distinguish between NAFLD and ALD [290]. ANI incorporates mean corpuscular volume (MCV), aminotransferase levels, body mass index (BMI), and sex (ANI = −58.5 + 0.637(MCV) + 3.91(AST/ALT) − 0.406(BMI) + 6.35 for men). An ANI greater than 0 or less than 0 favors a diagnosis of ALD or NAFLD, respectively. The probability of the patient having ALD rather than NAFLD is then calculated using the value obtained for the ANI as Probability = eANI/(1 + eANI).

## 11. ALD, NAFLD, and HCC

Both ALD and NAFLD share pathways that contribute to the spectrum of disease ranging from steatosis to steatohepatitis, cirrhosis, and HCC. The majority of primary liver tumors are HCC. HCC is a primary liver malignant tumor that typically develops in the setting of chronic liver disease. Major contributing conditions to HCC are liver cirrhosis or chronic hepatitis B virus infection. With the increasing prevalence of metabolic risk factors worldwide (metabolic syndrome, obesity, type II diabetes, and NAFLD), the risk of HCC will also increase globally. Furthermore, excessive alcohol consumption also remains an intractable risk factor for HCC [190,199,291]. Evidence confirms that alcohol intake increases the risk of liver cirrhosis and HCC, although the ultimate dose and duration of use remain poorly investigated [292,293,294]. A cohort study enrolled 652 patients with biopsy-confirmed alcoholic cirrhosis followed up for a median of 29 months. HCC developed in 7% of cases, equaling an estimated incidence of 2.9 cases per 100 patient-years [295]. Alcohol might induce either a direct toxic effect, an indirect effect via the development of liver cirrhosis [296], or synergistic effects in combination with another risk factor for HCC such as viral hepatitis [297].

The synergistic effect of alcohol can also occur in combination with metabolic factors such as diabetes mellitus [298] and/or obesity [299,300,301] and/or NAFLD.

Diabetes mellitus is a risk factor for HCC [300,302,303,304,305,306,307,308,309,310,311,312]. A large systematic review included 49 case-control and cohort studies; patients with diabetes mellitus had 2.2-fold increased HCC risk [302]. Similar results emerged from a meta-analysis of 14 prospective epidemiologic cases [312] and a population-based cohort study confirmed the findings on 19,349 patients with newly diagnosed diabetes and 77,396 patients without diabetes [306]. The incidence of HCC was higher among patients with diabetes compared with those without diabetes (21.0 vs. 10.4 per 10,000 person-years). Whether diabetes mellitus is a true associated factor for HCC requires further studies, since HCC precursors might rather be liver cirrhosis occurring before diabetes mellitus or NAFLD occurring in many diabetic patients. Another piece of evidence is that therapy with metformin is associated with decreased HCC risk [313,314,315,316]. As shown by a meta-analysis of eight observational studies, the risk for HCC in diabetic patients on metformin therapy could be as low as OR 0.50 (95% CI 0.34–0.73) [317].

Obesity is independently associated with liver cancer. In addition, obesity is often associated with diabetes mellitus and NAFLD [243,300,318,319,320,321].

Within the spectrum of NAFLD, NASH-related cirrhosis is a risk factor for HCC in Western countries [46,322,323,324,325]. The estimated annual incidence rate of HCC in patients with NASH cirrhosis was 1–2% [326]. In a large cohort study of patients with NASH cirrhosis, the incidence of HCC was 1 per 100 person-years of follow-up [327]. Although possible, the occurrence of HCC in patients with NAFLD but without cirrhosis is low [46,327,328], with an incidence as low as 0.008 per 100 person-years of follow-up [46].

## 12. Conclusions

NAFLD and ALD are nosologically considered two distinct entities, which in the population are major causes of nonviral liver cirrhosis. Although the orthodox dichotomic classification implies the existence of metabolic abnormalities, obesity, metabolic syndrome, and no or minimal alcohol consumption in NAFLD patients and 5–10 daily drinks for years in ALD [209], the two entities show a continuum and rather exert synergistic effects on the progression toward liver cirrhosis (Figure 2). The current view is that low alcohol intake increases the risk of advanced clinical liver disease in NAFLD, whereas metabolic factors increase the risk of cirrhosis among alcohol risk drinkers. Special attention should be given to individuals with metabolic abnormalities who consume small amounts of alcohol.

## Figures and Tables

**Figure 1 ijms-23-02636-f001:**
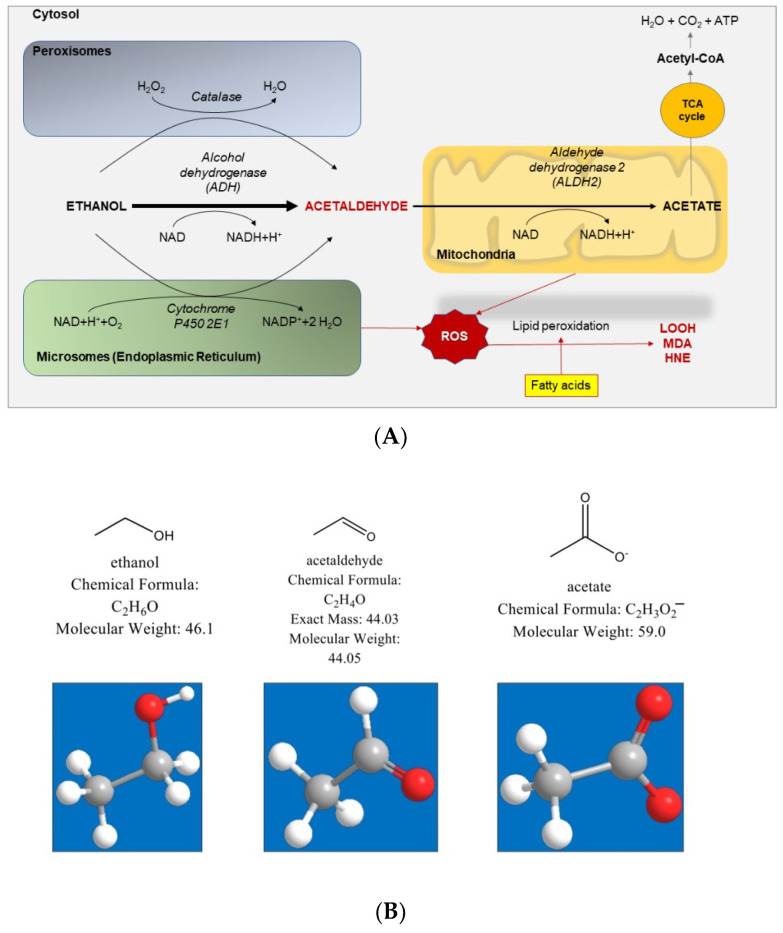
(**A**) Ethanol metabolism pathways in the liver. Ethanol is oxidized in the liver through three distinct pathways including (1) alcohol dehydrogenase (ADH) in the cytosol, ADH is the NAD+ dependent abundant enzyme responsible for the majority of ethanol metabolism; (2) cytochrome P450 2E1 (CYP2E1) in microsomes, this enzyme is NADPH cofactor-dependent belonging to the microsomal ethanol oxidizing system (MEOS); and (3) catalase in cell bodies peroxisomes, each enzymatic pathway produces acetaldehyde, a highly toxic and carcinogenic metabolite that forms adducts with DNA, lipids, and proteins. Acetaldehyde is detoxified to acetate by aldehyde dehydrogenase 2 (ALDH2) in mitochondria to form acetate and NADH. ROS, reactive oxygen species. TCA, tricarboxylic acid; HNE: 4-hydroxy-2-nonenal; LOOH: lipid hydroperoxides; MDA: malondialdehyde. The ROS pathway develops with excess alcohol consumption [104,105]. (**B**) Chemical formula, molecular weight, and 3D structure of ethanol, acetaldehyde, and acetate (https://pubchem.ncbi.nlm.nih.gov/, accessed on 8 February 2022).

**Figure 2 ijms-23-02636-f002:**
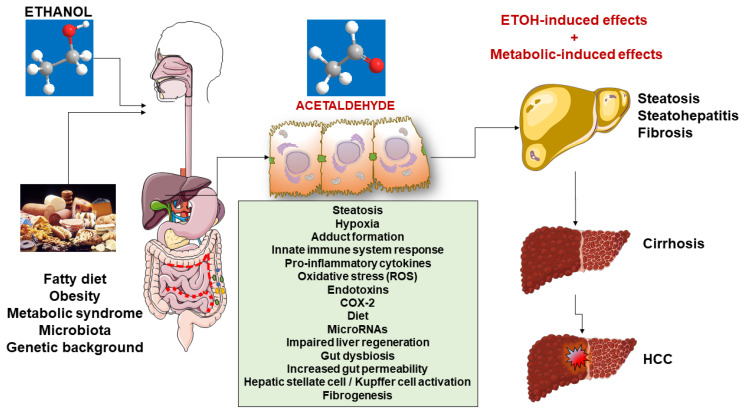
Synergistic and detrimental effects of ethanol and metabolic factor in the progression of liver disease. Abbreviations: COX-2, cycloxygenase-2; ETOH, ethanol; HCC, hepatocellular carcinoma; ROS, reactive oxygen species.

**Table 1 ijms-23-02636-t001:** Evidence of putative links between alcoholic liver disease (ALD) and nonalcoholic fatty liver (NAFLD).

Author	Setting	Experiment	Notes
Vecchione et al. [233]	FaO hepatoma cell culture	Incubation of cells with 0.35 mM free fatty acids oleate/palmitate alone, in combination with 100 mM ethanol, or ethanol alone.	Exposure to either fatty acids or ethanol led to steatosis. Effect was augmented in combination: (i) increased the lipid droplets number, although reducing their size; (ii) upregulated PPARγ and SREBP-1c and downregulated sirtuin-1 (SIRT1); (iii) impaired fatty acid oxidation; (iv) no change in lipid secretion and oxidative stress.
Cope et al. [234]	Mice model	To determine if the intestinal production of ethanol is increased in obesity. Breath collected from genetically obese, ob/ob male C57BL/6 mice and lean male littermates at different ages (14, 20, and 24 weeks) and times of the day (9 a.m., 3 p.m., and 9 p.m.). Obese mice (24 weeks old) were then treated with neomycin (1 mg/mL) for 5 days, and sampling was repeated.	Ethanol can be detected in exhaled breath. In obesity, an age-related increase in breath ethanol content reflects increased production of ethanol by the intestinal microflora. Intestinal production of ethanol may contribute to the genesis of obesity-related fatty liver.
Nagata et al. [222]	Animal models	Review of mechanisms of damage in ALD/NAFLD	Increased gut permeability. Gut-derived endotoxins. Activation of Kupffer cells. Activation of Toll-like receptor (TLR-4). Ethanol damage with cytokine production. Induction of the cytochrome P450 2E1 (CYP2E1) form of cytochrome P450 enzymes and TNF-alpha by ethanol. Common pathways in ALD/NAFLD Diabetes and obesity can induce CYP2E1 (increased ethanol degradation through the CYP2E1 pathway) with amplification of alcohol hepatotoxicity.
Xu et al. [235]	Mice	170% overnutrition in calories (intragastric overfeeding of high fat diet). Alcohol (low or high dose) was then co-administrated.	Moderate obesity (28–35% weight increase) plus alcohol intake causes synergistic steatohepatitis in an alcohol dose-dependent manner. Involvement of macrophagic activation, mitochondrial damage. Likely mechanisms: Nitrosative stress mediated by M1 macrophagic activation (innate immunity), adiponectin resistance, and accentuated endoplasmic reticulum and mitochondrial stress underlie potential mechanisms for synergistic steatohepatitis caused by moderate obesity and alcohol.
Grasselli et al. [236]	Rats	Effects produced by binge ethanol consumption in the liver of male Wistar rats fed a standard (Ctrl) or a high-fat diet HFD.	Double insult of diet and ethanol. Larger increase in fat accumulation within ADRP-positive lipid droplets. Stimulation of lipid oxidation in the attempt to limit excess fat accumulation. Induction of antioxidant proteins (MT2, in particular) to protect the liver from the ethanol-induced overproduction of oxygen radicals. Effect of diet and ethanol on lipid dysmetabolism might be mediated, at least in part, by PPARs and cytochromes CYP4A1 and CYP2E1.
de Medeiros et al. [237]	Animal models/human context	Review of mechanisms leading to increased production of endogenous ethanol in NAFLD.	Microbiota produces ethanol as a prodrug turning to acetaldehyde with hepatotoxic properties. According to the authors’ calculations, endogenous ethanol production may exceed 480 g daily in NAFLD patients. All genes involved in endogenous ethanol metabolism are upregulated in the livers of patients with nonalcoholic steatohepatitis (NASH). Overexpression of the gene encoding alcohol dehydrogenase (ADH) 4 implicates liver exposure to high concentrations of endogenous ethanol. NAFLD might represent a model of endogenous alcoholic fatty liver disease (EAFLD).
Guo et al. [238]	Rodents	Lieber–De Carli liquid diet. Induction of alcoholic liver injury with addition of fat components	Double ethanol/metabolic damage.
Minato et al. [239]	Rat model	(a) Thirty-week-old male Otsuka Long-Evans Tokushima fatty (OLETF) and (b) control, male Otsuka Long-Evans Tokushima (OLET). Oral administration of 10 mL of 10% ethanol orally for 5, 3, 2, and 1 d/wk for 3 consecutive weeks. Assessment of various biochemical parameters of obesity, steatosis and NASH were monitored in serum and liver specimens in untreated and ethanol-treated rats. The liver sections were evaluated for histopathological alterations of NASH and stained for cytochrome P-4502E1 (CYP2E1) and 4-hydroxy-nonenal (4-HNE).	Simple steatosis, hyperinsulinemia, hyperglycemia, insulin resistance, hypertriglycemia, and marked increases in hepatic CYP2E1 and 4-HNE were present in 30-wk-old untreated OLETF rats. Massive steatohepatitis with hepatocyte ballooning in the livers of all OLETF rats treated with ethanol. Serum and hepatic triglyceride levels as well as tumor necrosis factor (TNF)-alpha mRNA were markedly increased in all ethanol-treated OLETF rats. Marked increases in the hepatic tissue of all the groups of OLETF rats treated with ethanol compared with OLET rats. Findings suggest that “a binge” serves as a “second hit” for development of NASH from obesity-induced simple steatosis through aggravation of oxidative stress.
Duly et al. [240]	Mice model	C57BL6 male mice fed either chow or high-fat diet (HFD) ad libitum for 12 weeks. A sub-set of mice from each group were also given alcohol (2 g kg(-)(1) body weight) twice a week via intra-gastric lavage.	HFD induced hepatic steatosis. HFD significantly increased total body weight, triglyceride and cholesterol, whereas alcohol increased liver weight. Alcohol+HFD in combination produced maximum hepatic steatosis, increased micro- and macro-vesicular lipid droplets, increased de novo lipogenesis (steroid response-element binding protein 1 (SREBP-1) and stearoyl-CoA desaturase-1 (SCD-1)) and proliferation peroxisome activated receptor alpha (PPARα), and decreased fatty acid beta-oxidation (Acyl-CoA oxidase 1 (ACOX1)). Alcohol+HFD treatment also increased the inflammation (CD45+, CD68+, F4/80+ cells; tumour necrosis factor-alpha (TNF-alpha), F4/80 mRNAs) and fibrogenesis (vimentin+ activated stellate cells, collagen 1 (Col1) production, transforming growth factor-beta (TGF-beta) and Col-1 mRNAs) in mice livers. This is a novel mouse model with more severe liver injury than either alcohol or HFD alone recapitulating the human setting of intermittent alcohol drinking and HFD.
Baker et al. [241]	Human model	To test the expression of inflammation, fibrosis, and alcohol metabolism-related genes in the liver tissues of NASH patients and normal controls. Microarray and quantitative real-time PCR.	Small number of cases. Genes related to liver inflammation and fibrosis were elevated in NASH livers compared to normal livers. Increased gene transcription of alcohol dehydrogenase (ADH) genes, genes for catalase and cytochrome P450 2E1, and aldehyde dehydrogenase genes. First human evidence that suggests endogenous alcohol may contribute to the development of NAFLD.
Zhu et al. [116]	Human model	Composition of gut bacterial communities of NASH, obese, and healthy children (16S ribosomal RNA pyrosequencing). Peripheral blood ethanol levels assessed as marker of endogenous ethanol production of patients and healthy controls.	Most of the microbiome samples clustered by disease status. Increased abundance of alcohol-producing bacteria in NASH microbiomes. Proteobacteria, Enterobacteriaceae, and Escherichia were the only phylum, family, and genus types exhibiting significant difference between obese and NASH microbiomes. Similar blood-ethanol concentrations were observed between healthy subjects and obese non-NASH patients NASH patients exhibited significantly elevated blood ethanol levels.
Parker et al. [205]	Human model	Review on the effect of alcohol on adiposity and adipose tissue and the relationship between alcohol, adipose tissue, and the liver.	Alcohol can disrupt extrahepatic fat tissue function and cause adipocyte death with subsequent proinflammatory responses and increased lipolysis. Factors contribute to liver damage by indirect mechanisms. A high-fat diet sensitizes adipose tissue to alcohol-induced lipolysis.
Aragonès et al. [117]	Human model	Measurement of circulating microbiota-derived metabolites from women with normal weight, morbid obesity, with or without NAFLD. Liver biopsy performed to differentiate between simple steatosis and steatohepatitis. Measurement of choline and its derivatives, betaine, endogenous ethanol, bile acids, short-chain fatty acids, and soluble Toll-like receptors.	Endogenous circulating ethanol levels were increased in NASH patients in comparison to those with simple steatosis.

**Table 2 ijms-23-02636-t002:** Major pitfalls in the studies linking ethanol consumption with obesity vs. liver damage [253].

Incomplete study design
Unclear endpoints
No adjustment for dietary factors, physical exercise, smoking, coffee consumption, or economic and social aspects
Lack of proper stratification for ethnicity
Poor information about comorbidities
Insufficient information about pattern and type of alcohol use, lifetime alcohol intake (more than average alcohol intake), distinction between lifetime abstainers vs. current abstainers (which might have included former heavy drinkers)
Underreporting alcohol use

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
