# Peer review of "Synergistic and Detrimental Effects of Alcohol Intake on Progression of Liver Steatosis"

_ijms, 2022, doi:10.3390/ijms23052636_

Round 1

Reviewer 1 Report

The overlap between ALD and NAFLD regarding the histopathology and pathogenesis is an important area of growing interest.  This is a timely review of the epidemiological and clinical features of NAFLD/ALD. 

  1. Although the review is generally well written, there are some errors/typos that need to be corrected, such as errors in line 28, 59, 294, 298, 354, 410, etc.
  2. Some of the writings need to be more accurate and informative. For example, in the paragraph of line 392-394, it says "Another down-regulated microRNA is microRNA-122, which implies that microRNA-182, the microRNA described before miR-122, is also down-regulated. However, microRNA-182 is actually induced in alcoholic hepatitis.  There are some other cases that the description is not very accurate or informative.  When "changes" or "association" was used, it would be more informative to tell the reader whether it is increased/decreased, or positive/negatively associated.
  3. In the paragraph of line 711-718, the description on ANI is not clear, please rewrite.

Author Response

Comments and Suggestions for Authors

The overlap between ALD and NAFLD regarding the histopathology and pathogenesis is an important area of growing interest.  This is a timely review of the epidemiological and clinical features of NAFLD/ALD. 

  1. Although the review is generally well written, there are some errors/typos that need to be corrected, such as errors in line 28, 59, 294, 298, 354, 410, etc.

All errors/typos have been corrected following a careful revision of the manuscript. Most relevant changes are marked in red text

  1. Some of the writings need to be more accurate and informative. For example, in the paragraph of line 392-394, it says "Another down-regulated microRNA is microRNA-122, which implies that microRNA-182, the microRNA described before miR-122, is also down-regulated. However, microRNA-182 is actually induced in alcoholic hepatitis.  There are some other cases that the description is not very accurate or informative.  When "changes" or "association" was used, it would be more informative to tell the reader whether it is increased/decreased, or positive/negatively associated.

We thank the reviewer for this comment.

The manuscript was checked for a better description of reported associations.

The cited paragraph has been now changed as follows:

MicroRNAs are non-coding RNA species acting through epigenetic mechanisms. MicroRNA-182 is the most highly expressed miRNA in alcoholic hepatitis. MicroRNA-182 expression is associated with disease severity, short-term mortality, and the ductular reaction of alcoholic hepatitis [174]. Conversely, the microRNA-122 is down-regulated in patients with alcoholic hepatitis and is also involved in the progression of the disease [175].

  1. In the paragraph of line 711-718, the description on ANI is not clear, please rewrite.

The paragraph has been now rewritten as follows:

The ALD/NAFLD index (ANI), a specific algorithm generated by logistic regression models, is a predictive model to distinguish between NAFLD and ALD [299]. ANI incorporates mean corpuscular volume (MCV), aminotransferase levels, body mass index (BMI), and sex (ANI = -58.5 + 0.637 (MCV) + 3.91 (AST/ALT) – 0.406 (BMI) + 6.35 for men). An ANI greater than 0 or less than 0 favors a diagnosis of ALD or NAFLD, respectively. The probability of the patient having ALD rather than NAFLD is then calculated using the value obtained for the ANI as Probability = eANI/(1+eANI).

Reviewer 2 Report

The authors of the article are thanked for the quality of the work and the choice of the subject relating to the effects of alkohol intake and metabolic disordres on fatty liver. It is known that  nonalcoholic fatty liver disease (NAFLD) and alcoholic liver disease (ALD) are the most common liver disorders worldwide. Agostino Di Ciaulaet and collaborators discuss the epidemiological and clinical features of NAFLD/ALD, aspects of alcohol metabolism and mechanisms of damage with respect to steatosis, fibrosis, cumulative effects, and deleterious consequences which include hepatocellular carcinoma. The manuscript entitled „Synergistic and detrimental effects of alcohol intake and metabolic disorders on fatty liver”  by Agostino Di Ciaulaet al, is a good study, well executed, and deserve some space in the journal. Only three minor concerns have been raised:

  • Page 6, line 234: the statement „ROS, reactive oxygen species” had already appeared in the line 232. If this is a repetition, please delete it.
  • Page 6, line 235: the statement „TCA, tricarboxylic acid” had already appeared in the lines 232-233. If this is a repetition, please delete it.
  • Page 6, lines 235-236: „Chemical formula, molecular weight and 3D structure of etha-235 nol, acetaldehyde, and acetate”. Please add a reference for this statement.

Author Response

Reviewer 2

Comments and Suggestions for Authors

The authors of the article are thanked for the quality of the work and the choice of the subject relating to the effects of alkohol intake and metabolic disordres on fatty liver. It is known that  nonalcoholic fatty liver disease (NAFLD) and alcoholic liver disease (ALD) are the most common liver disorders worldwide. Agostino Di Ciaulaet and collaborators discuss the epidemiological and clinical features of NAFLD/ALD, aspects of alcohol metabolism and mechanisms of damage with respect to steatosis, fibrosis, cumulative effects, and deleterious consequences which include hepatocellular carcinoma. The manuscript entitled „Synergistic and detrimental effects of alcohol intake and metabolic disorders on fatty liver”  by Agostino Di Ciaulaet al, is a good study, well executed, and deserve some space in the journal. Only three minor concerns have been raised:

  • Page 6, line 234: the statement „ROS, reactive oxygen species” had already appeared in the line 232. If this is a repetition, please delete it.

Thank you for this observation. The repetition has been now deleted from the text.

  • Page 6, line 235: the statement „TCA, tricarboxylic acid” had already appeared in the lines 232-233. If this is a repetition, please delete it.

The repetition has been now deleted from the text.

  • Page 6, lines 235-236: „Chemical formula, molecular weight and 3D structure of etha-235 nol, acetaldehyde, and acetate”. Please add a reference for this statement.

The sentence has been now rewritten as follows:

“B) Chemical formula, molecular weight and 3D structure of ethanol, acetaldehyde, and acetate (https://pubchem.ncbi.nlm.nih.gov/, (accessed 8 February 2022).”